# Synergistic Hypoglycemic Effects of Pumpkin Polysaccharides and Puerarin on Type II Diabetes Mellitus Mice

**DOI:** 10.3390/molecules24050955

**Published:** 2019-03-08

**Authors:** Xue Chen, Lei Qian, Bujiang Wang, Zhijun Zhang, Han Liu, Yeni Zhang, Jinfu Liu

**Affiliations:** 1College of Food Science and Biotechnology, Tianjin Agricultural University, Tianjin 300384, China; tjchenx1993@sina.cn (X.C.); bujiangwang2018@163.com (B.W.); 13072253760@163.com (H.L.); 2Tianjin Research Institute of Forestry and Pomology, Tianjin Academy of Agricultural Sciences, Tianjin 300384, China; qianlei507@126.com (L.Q.); tjzhangzj@sina.cn (Z.Z.); 3Key Laboratory of Storage of Agro-products, Ministry of Agriculture, Tianjin 300384, China

**Keywords:** type II diabetes mellitus, pumpkin polysaccharides, puerarin, mechanism

## Abstract

To investigate the hypoglycemic effect and potential mechanism of pumpkin polysaccharides and puerarin on type II diabetes mellitus (T2DM) mice, mice were fed a high-fat diet and injected intraperitoneally with streptozotacin to induce T2DM. After eight weeks of drug administration, blood samples were withdrawn from tail veins of mice that had been fasted overnight. The results showed that both pumpkin polysaccharides and puerarin, as well as a pumpkin polysaccharides and puerarin combination, could ameliorate T2DM. The pumpkin polysaccharides and puerarin combination had a synergetic hypoglycemic effect on T2DM mice that was greater than the pumpkin polysaccharides’ or the puerarin’s hypoglycemic effect. Both the pumpkin polysaccharides and the puerarin were found to ameliorate the blood glucose tolerance and insulin resistance of T2DM mice. They showed lipid-lowering activity by reducing the total cholesterol, triglycerides, and low-density lipoprotein levels, and improving the high-density lipoprotein level. They had beneficial effects on the oxidative stress by decreasing the reactive oxygen species and malondialdehyde levels, and increasing the glutathione level and the superoxide dismutase activity. Furthermore, the nuclear factor E2 related factor 2 (Nrf2), heme oxygenase-1, and phosphoinositide-3-kinase (PI3K) levels were upregulated, and the Nrf2 and PI3K signalling pathways might be involved in the hypoglycemic mechanism. The combined administration of pumpkin polysaccharides and puerarin could synergistically ameliorate T2DM.

## 1. Introduction

Diabetes mellitus (DM) is a heterogeneous, multifactorial, chronic metabolic disorder characterized by hyperglycemia due to an insulin insufficiency or insulin resistance (IR) [1]. Insulin and oral hypoglycemic agents, including biguanides, thiazolidinediones, and sulfonylureas, can effectively control hyperglycemia. However, they also have prominent side effects, such as hypoglycemia and gastrointestinal disturbances [2]. Therefore, there is an urgent need to look for efficacious alternatives with fewer side effects.

Numerous studies have shown that active ingredients, including flavonoids, polyphenols, alkaloids, and polysaccharides, such as pumpkin, bean, oat, and balsam pear polysaccharides, exert beneficial effects on DM and its complications through protecting against oxidative stress damage or improving insulin sensitivity, and the combination of two or more kinds of ingredients can usually achieve better results through different targets [3,4,5,6,7].

Pumpkin has been used for the prevention of various diseases, such as hyperlipidemia and hypoglycemia [8,9,10]. It has been reported that pumpkin polysaccharides showed hypoglycemic activity in diabetic mice via increasing the plasma insulin levels [9], and protected islet cells from streptozotacin (STZ) injury via increasing the superoxide dismutase (SOD) levels [11,12].

Puerarin (Figure 1) is an isoflavonoid extracted from pueraria lobata roots, and has been used for various medicinal purposes in traditional oriental medicine. Previous reports have demonstrated that puerarin has a variety of biological actions in such diseases as osteoporosis, cardiovascular disease, and gynecological disease [13,14,15,16]. Modern pharmacological research has demonstrated that puerarin exerts a protective effect on the kidneys of diabetic rats [17,18,19].

Many researchers have devoted themselves to the study of pumpkin polysaccharides; so far, only acidic polysaccharides have been found. Based on the identified characteristics, the pumpkin polysaccharides that have been separated in our lab are novel. Although the anti-diabetic effects of puerarin and pumpkin polysaccharides in rodent diabetes models have already been described in multiple studies, their synergistic hypoglycemic effect has not been reported. To our knowledge, the mechanism that underlies the compatibility between pumpkin polysaccharides and puerarin has not been elucidated, which greatly restricts their coadministration and hinders us from realizing a greater hypoglycemic effect. In the present study, we intended to verify the synergistic hypoglycemic effect of pumpkin polysaccharide and puerarin coadministration on type II diabetes mellitus (T2DM) mice. Furthermore, we investigated the potential mechanism underlying the synergetic hypoglycemic effect of pumpkin polysaccharides and puerarin on T2DM mice. The results suggest that the antioxidant capacity of pumpkin polysaccharides and puerarin may play an important role in reducing the risk of T2DM through the upregulation of nuclear factor E2 related factor 2 (Nrf2)-mediated target genes and the phosphoinositide-3-kinase (PI3K) signalling pathway.

## 2. Results

### 2.1. Identification of Pumpkin Polysaccharides

#### 2.1.1. Determination of Molecular Weight

The molecular weight (Mw) of the pumpkin polysaccharides was evaluated by high-performance gel permeation chromatography (HPGPC) (Figure 2). Each fraction was represented by a broad and symmetrical peak in the chromatograms. The Mw, peak Mw, and average Mw was 749.3 kDa, 727.0 kDa, and 607.6 kDa, respectively. The polydispersity was 1.233. The polydispersity indicates a polymer’s molecular weight distribution, and the greater the polydispersity, the wider the molecular weight distribution. Generally, the polydispersity value of the polymer ranged from 1.5 to 2.0. The polydispersity of the pumpkin polysaccharides indicated that the molecular weight’s distribution range was narrow.

#### 2.1.2. Monosaccharide Composition Analysis

Pumpkin polysaccharides were hydrolyzed with trifluoroacetic acid, and the composition of the monosaccharides was analyzed by high-performance liquid chromatography (HPLC) (Figure 3). Compared with the retention time of the standard monosaccharide, the chromatogram showed that the pumpkin polysacchrides were composed of mannose (142.92 mg/g), ribose (42.89 mg/g), glucosamine (1.03 mg/g), glucuronic acid (17.83 mg/g), galacturonic acid (2.6 mg/g), glucose (125.75 mg/g), galactosamine (0.85 mg/g), xylose (112.34 mg/g), and fucose (73.25 mg/g).

#### 2.1.3. Ultraviolet Spectra

The purity of the pumpkin polysaccharides was confirmed by ultraviolet spectrophotometry. The absence of peaks between 260 nm and 280 nm indicated that the pumpkin polysaccharides were free of nucleic acid and proteins.

#### 2.1.4. Fourier-Transform Infrared Spectra

The infrared spectrum indicated that the pumpkin polysaccharides had the typical features of polysaccharides. There was an absorption peak at 3288.78 cm^−1^, which was mainly caused by the stretching vibration of the polysaccharide glycoside hydroxyl. There was a stretching vibration of an absorption peak C-H at 2911.10 cm^−1^. The absorption peak at 1645.94 cm^−1^ displayed the -CHO stretching vibration or the N-H deviational vibration. The absorption peak at 1417.53 cm^−1^ displayed the C-O stretching vibration. The absorption peaks between 1250 cm^−1^ and 950 cm^−1^ indicated that the glucose conformation was of the pyranose type.

### 2.2. Effect on Body Weight and Water Intake

Body weight was recorded every four days, and the growth curve is shown in Figure 4a. After eight weeks, there was a significant decrease in the body weight of the mice in the DM group (34.3 ± 2.3 g) with respect to that in the control (C) group (45.5 ± 3.1 g) (*p* < 0.05). The mice in the pumpkin polysaccharides (PP) group, the puerarin (P) group, and the pumpkin polysaccharides and puerarin (PPP) group showed a significant increase in body weight with respect to that in the DM group (*p* < 0.05).

As shown in Figure 4b, the water intake of the mice in the C group consistently maintained a low level during the experimental process. After eight weeks, the water consumption of the mice in the DM group (38.5 ± 2.6 mL) was much higher than that in the C group (17.1 ± 1.3 mL) (*p* < 0.05). After treatment with PP, P, and PPP, the mice in these groups significantly reduced their water consumption with respect to that in the DM group (*p* < 0.05).

### 2.3. Oral Glucose Tolerance Test

An oral glucose tolerance test (OGTT) can be used to diagnose diabetes or prediabetes. As shown in Figure 5, the glucose level of the mice in the C group reached its highest value (15.3 ± 1.3 mM) after 30 min and then gradually fell to the initial level. Compared with the C group, there was a significant increase in the glucose level of the mice in the DM group (32.7 ± 2.1 mM) (*p* < 0.05). Although the PP, P, and PPP groups showed a significant increase in the glucose level after 30 min, the mice in these groups efficiently suppressed the peak value after 60 min, and the glucose level in the PPP group returned approximately to the baseline after 120 min.

The area under the curve (AUC) value indicates the potency of blood glucose tolerance. As shown in Figure 6, after eight weeks, the blood glucose response in the C group (23.5 ± 1.7) was significantly reduced with respect to that in the DM group (57.3 ± 4.5), which was similar to the result of lowering the glucose level. Compared with the DM group, the mice in the PP, P, and PPP groups had a lower AUC value (*p* < 0.05). The mice in the PPP group had a more potent glucose tolerance than the mice in the P and PP groups. These results indicate that pumpkin polysaccharides and puerarin are potent in the treatment of diabetes and that their combined effect is greater.

### 2.4. Blood Glucose and Insulin Level

As shown in Table 1, after eight weeks, there was a significant increase in the serum glucose level of the mice in the DM group (17.85 ± 1.53 mM) in comparison with that in the C group (3.03 ± 0.27 mM) (*p* < 0.05). After treatment with PP, P, and PPP, the mice in these groups showed a significant decrease in the serum glucose level (*p* < 0.05), which reached 9.87 ± 1.07, 11.23 ± 1.28, and 6.35 ± 0.53 mM, respectively, in comparison with that in the DM group.

A similar result was also observed in the serum insulin level. Compared with the serum insulin level (13.05 ± 0.95 mU/L) of the mice in the DM group, pumpkin polysaccharides or puerarin administration led to a significant decrease (8.37 ± 0.75 and 8.03 ± 0.66 mU/L, respectively) in the serum insulin level, and the effect on the combined group was greater.

After eight weeks, the IR level of the mice in the DM group (10.77 ± 1.05) was much higher than that in the C group (1.11 ± 0.09) (*p* < 0.05). After treatment with PP, P, and PPP, the mice in these groups had a lower IR value, which reached 3.75 ± 0.35, 4.05 ± 0.47, and 2.53 ± 0.23, respectively, in comparison with that in the DM group. These results proved that pumpkin polysaccharides and puerarin could ameliorate the insulin resistance caused by T2DM.

### 2.5. Effect on Serum Lipid Levels

As shown in Table 2, after eight weeks, significant changes in the serum lipid levels were observed in the DM group in comparison with that in the C group. STZ induction resulted in a significant increase in the serum total cholesterol (TC), triglycerides (TG), and low-density lipoprotein (LDL) levels, which reached 5.28 ± 0.49, 2.75 ± 0.23, and 2.79 ± 0.23 mM, respectively (*p* < 0.05), while a significant reduction in the serum high-density lipoprotein (HDL) level (0.83 ± 0.07 mM) was observed (*p* < 0.05). Compared with the DM group, the administration of pumpkin polysaccharides or puerarin not only significantly reduced the serum TC, TG, and LDL levels, but also improved the serum HDL level, and the effect on the combined group was greater. Consistent changes in free fatty acids (FFAs) were also observed. These results indicate that pumpkin polysaccharides and puerarin have lipid-lowering activity.

### 2.6. Effect on Oxidative Stress

To evaluate the antioxidant capacity, the reactive oxygen species (ROS), malondialdehyde (MDA), glutathione (GSH), and SOD activity levels were determined after eight weeks. As shown in Table 3, the mice in the DM group were under oxidative stress with the amount of ROS (597.1 ± 25.01 IU/L) and MDA (11.59 ± 0.95 nmol/L) that was generated. The GSH level and SOD activity in the DM group decreased significantly (38.7% and 42.8%, respectively) in comparison with that in the C group (*p* < 0.05). After treatment with PP, P, and PPP, the mice in these groups had a higher GSH level and SOD activity than that in the DM group (*p* < 0.05). Furthermore, the MDA and ROS levels in the treated groups decreased significantly (*p* < 0.05). Hereby, by blocking the increase of MDA and ROS levels associated with an elevation in antioxidant ability, including the GSH level and the SOD activity, both pumpkin polysaccharides and puerarin had beneficial effects on the oxidative stress, and their synergetic effect was much greater.

### 2.7. Western Blot Analysis

Nrf2 is a key transcription factor for combating hepatic oxidative stress and controls antioxidant response element (ARE)-dependent gene regulation [20,21]. Nrf2 sequestered in the cytoplasm by cytosolic repressor Kelch-like ECH-associated protein 1 (Keap 1) plays an important role in the maintenance of the cellular redox balance [22]. Oxidative stress facilitates Nrf2’s escape from Keap1-mediated degradation and subsequent nuclear translocation [23]. The results displayed in Figure 7a indicate that, after eight weeks, the ratio of Nrf2/LaminB1 (nucleus) in the DM group was significantly decreased in comparison with that in the C group (*p* < 0.05). After treatment with PP, P, and PPP, the level of Nrf2 (nucleus) was notably increased (*p* < 0.05). With respect to Nrf2 expression in the liver, the PPP group had a more efficient effect than the PP group or the P group. The increased expression of Nrf2 would be helpful for the activation of associated proteins.

Phase II antioxidant enzymes are regulated by AREs at the transcription level. Heme oxygenase-1 (HO-1) is a rate-limiting enzyme involved in the conversion of heme to biliverdin and carbon monoxide, among other cellular reactions [24]. As shown in Figure 7b, after eight weeks, the expression of HO-1 in the DM group was obviously decreased in comparison with that in the C group (*p* < 0.05). After treatment with PP, P, and PPP, the levels of HO-1 in these groups were notably increased compared with that in the DM group (*p* < 0.05). With respect to HO-1 expression in the liver, the PPP group had a more efficient effect than the PP group or the P group. These results indicate that pumpkin polysaccharides and puerarin might be potential therapeutic agents through their activation of the Nrf2/HO-1 signalling pathway.

The PI3K/Akt pathway is a key component of the insulin signaling cascade and considered to be necessary for glucose transport [25,26]. As shown in Figure 8, after eight weeks, the phosphorylation of PI3K and Akt in the hepatic tissue was significantly decreased the in DM group (*p* < 0.05). Treatment with PP, P, and PPP significantly increased the fraction of phosphorylated PI3K and partially increased the fraction of phosphorylated Akt in the hepatic tissue. With respect to the phosphorylation of Akt in the hepatic tissue, the PPP group had a more efficient effect than the PP group or the P group. These findings suggest that the activation of the PI3K/Akt signal pathway may contribute to the effects that pumpkin polysaccharides and puerarin have as demonstrated in the present study.

## 3. Discussion

STZ is often used to induce DM in experimental animals through the selective destruction of insulin-producing pancreatic endocrine cells [27]. STZ causes the alkylation of DNA and leads to β-cells’ destruction [28]. However, a high dose of STZ is likely to induce a short period of prediabetes conditions. Recently, several animal models have been employed in diabetes research, out of which the HFD regimen was utilized to induce T2DM. This approach mimics the availability of a fat-rich diet in our modern society, thereby contributing to the cause of diabetes in humans [29,30,31]. However, the induction period of HFD is too long [31]. In this study, the animal model for T2DM was induced by feeding the animals with a HFD following a low dose of STZ.

Pumpkin is rich in polysaccharides, carotene, minerals, and vitamins that are beneficial to health [32,33]. Recently, pumpkin polysaccharides have attracted more attention because they have many biological effects, such as detoxification, antioxidation, blood pressure reduction, and blood lipid reduction effects [10,34]. Polysaccharides show strong activities in experimental animals, and could also act as delivery-functional materials with other constituents [35,36]. The combination of polysaccharides and other plant-derived constituents could enhance their pharmacological activity and decrease their toxicity [37]. During a screening of natural products that enhance the effect of pumpkin polysaccharides on glucose control, we found that puerarin exhibits good synergistic activity against hyperglycaemia, suggesting that this combination might be useful in prediabetes treatment.

Weight loss is a main symptom of diabetes, and it could be due to a loss of appetite, increased muscle waste, and a loss of tissue proteins [38]. In this study, the treatment groups were found to maintain the body weight of diabetic mice. In addition, the administration of the drug led to a decrease in water consumption, indicating that pumpkin polysacchrides and puerarin can regulate the water balance.

This study showed that pumpkin polysacchrides and puerarin significantly diminished the blood glucose levels in diabetic mice. Both pumpkin poloysacchrides and puerarin are extracts from natural plants, and this hypoglycemic effect may be explained in part by either a decrease in the rate of intestinal glucose absorption or an increase in peripheral glucose utilization.

HFD feeding and STZ induction trigger extra TG and TC accumulation in mice livers, representing an imbalance between the complex interactions of metabolic events [39]. Consistent with a change in serum lipid levels, all treatment groups showed a significant beneficial change in liver function and lipid deposits. We found that treatment with the drug lowered the serum TC, TG, and LDL levels and elevated the serum HDL level. This suggests that pumpkin polysaccharides and puerarin may help to decrease the incidence of cardiovascular disease. In the liver, an increase in FFAs contributes to resistance to the action of insulin by stimulating endogenous glucose production [40,41].

Chronically raised FFA levels have a lipotoxic effect on the pancreas, and lipid accumulation in beta cells might lead to reductions in insulin secretion [42,43]. In this study, treatment with the drug was found to reduce FFA levels to normal, implying that pumpkin polysaccharides and puerarin may work synergistically to prevent excess lipid accumulation in the liver and favour glucose homeostasis.

In a hyperglycemic state, there is an increase in oxidative stress, which causes a defect in insulin action and secretion [44]. Several studies have documented the influence of free radicals on blood glucose, lipid peroxidation, and low-density lipoprotein in diabetes progression [45]. Oxygen free radicals exert cytotoxic effects on membrane phospholipids, resulting in the formation of MDA. As a product of lipid peroxidation, the level of MDA reflects the degree of oxidation in the body. Antioxidant enzymes play important protective roles in the pathogenesis of DM. The superoxide radical is converted to H_2_O_2_ by a group of enzymes known as SOD. GSH can reduce H_2_O_2_, hydroperoxides, and xenobiotic toxicity, and can also be involved in the enzymatic detoxification reaction for ROS [46]. In this study, HFD feeding and STZ induced oxidative stress by inhibiting the activities of antioxidant enzymes. Pumpkin polysaccharides and puerarin could protect liver cells against oxidative injury through increasing SOD and GSH activities and decreasing the MDA content. They were found to significantly decrease an elevated ROS level, and were able to regulate the balance of oxygenation effectively in the diabetic mice.

We investigated the potential mechanisms underlying the metabolic regulation through an examination of the key proteins that control oxidative stress. Nrf2 mediates a broad-based set of adaptive responses to intrinsic and extrinsic cellular stresses. Nrf2’s regulation of cellular antioxidant and anti-inflammatory machinery plays a central part in defense against oxidative stress, with a particular role in the regulation of phase II detoxifying enzymes and the antioxidant status [47,48]. In the presence of oxidative stress, covalent modification of cystein residues in the Keap1 molecule disables its ability to bind Nrf2, thereby promoting its translocation to the nucleus. In the nucleus, Nrf2 binds to ARE on its target genes, promoting the transcription of phase II enzymes, including HO-1, GPx, SOD, and CAT [49,50,51]. These target genes are upregulated through binding of Nrf2 to ARE found in the promoters of these genes. Strategies aimed at enhancing endogenous antioxidant defense systems by restoring Nrf2 activity are more effective in the management of oxidative stress, as shown in this study. Our data demonstrated a significant increase in nucleus Nrf2 and HO-1 expression upon treatment with pumpkin polysaccharides and puerarin.

The PI3K/Akt signalling pathway plays a role in many essential cellular physiological processes, such as survival, proliferation, migration, and differentiation, which can be activated by extracellular stimuli, such as cytokines and growth factors. Through the PI3K/Akt pathway, ROS can activate the Nrf2/ARE antioxidant pathway to protect the cells [26,52]. The PI3K/Akt pathway regulates HO-1 expression, and Nrf2 transcriptional activation and PI3K/Akt pathways are involved in the phosphorylation of Nrf2 to facilitate disassociation with Keap1 and nuclear translocation [53,54]. Thus, PI3K is an important regulator of cell growth and survival in response to oxidative stress [54]. Our study suggests that the PI3K/Akt signalling pathway may be involved in the induction of Nrf2/ARE-driven gene expression. Meanwhile, the PI3K/Akt signaling pathway is a classic insulin pathway because it plays a role in glucose uptake by the liver, skeletal muscles, and adipose tissues [25]. Decreasing or blocking this pathway can reduce the physiological effects of insulin, which may lead to IR. Activated Akt transmits the signal to the downstream receptor substrate and produces various biological effects [55,56], such as glucose absorption, glycolysis, glycogen synthesis, and protein synthesis. Treatment with pumpkin polysaccharides and puerarin was observed to increase the gene and protein expressions of PI3K and Akt in the livers of diabetic mice. These results suggest that pumpkin polysaccharides and puerain have hypoglycemic properties and may alleviate insulin resistance in T2DM mice through the PI3K/Akt pathway. This research indicates that the hypoglycemic effect of the treatment with the pumpkin polysaccharides and puerarin combination was greater than that of the treatment with either pumpkin polysaccharides or puerarin. However, the mechanism of action of the active compounds in both extracts needs further investigation.

## 4. Materials and Methods

### 4.1. Materials and Reagents

Pumpkin polysaccharides were extracted from pumpkin and then separated by a DEAE-52 cellulose column and Sephadex G-100 column chromatography, successively. Puerarin with a purity of over 98% was purchased from Sigma Chemical Co. (St. Louis, MO, USA).

Streptozotocin (STZ) was purchased from Sigma Chemical Co. (St. Louis, MO, USA). Primary antibodies for Nrf2, HO-1, PI3K, Akt, β-actin, and Lamin B1 were purchased from Abcam (Cambridge, MA, USA). Nuclear/Cytosol isolation kits were purchased from the Beyotime Institute of Biotechnology (Beijing, China). Detection kits for the ROS, MDA, and GSH levels and the SOD activity were purchased from the Jiancheng Bioengineering Institute (Nanjing, China). Insulin ELISA kits and the secondary antibodies peroxidase-conjugated goat anti-rabbit lgG and peroxidase-conjugated goat anti-mouse lgG were purchased from Solar Biotechnology Co., Ltd. (Beijing, China). All other chemicals were of analytical grade and purchased from local firms.

### 4.2. Separation and Identification of Pumpkin Polysaccharides

#### 4.2.1. Separation of Pumpkin Polysaccharides

The polysaccharides from a pumpkin powder were extracted with water at 80 °C for 2 h, following treatment with chloroform and butanol at a ratio of 4:1. Then, the mixture was concentrated and precipitated with three volumes of ice-cold ethyl alcohol at 4 °C overnight. The precipitate was collected by centrifugation and lyophilized to obtain a brownish powder. The crude precipitate, with a high molecular weight component of water-soluble substances, was separated by using DEAE-cellulose anion-exchange chromatography. Each polysaccharide fraction was collected and analyzed by the phenol–sulfuric acid method. Then, the major fraction was loaded into a Sephadex G-100 gel column chromatography system for further purification. The purified polysaccharides were collected and freeze-dried for further analysis.

#### 4.2.2. Molecular Weight Determination

Molecular weight was evaluated by HPGPC (Wyatt Technology Corporation, Santa Barbara, CA, USA) [57]. Pumpkin polysaccharides and dextran standards were dissolved in deionized water at a concentration of 2.0 mg/mL and then analyzed on an Agilent 1100 series HPLC system to determine the retention time. The column and detector compartments were maintained at 30 °C and 35 °C, respectively. Distilled water was used in the mobile phase. The detection rate was 1.0 mL/min, and the tested volume was 10 µL.

#### 4.2.3. Monosaccharide Composition Analysis

Pumpkin polysacchrides (10 mg) were hydrolyzed in 2 M trifluoroacetic acid at 100 °C for 6 h. The residual acid was removed with methyl alcohol. The hydrolyzates were dried and dissolved with distilled water, and then analyzed by HPLC on an Agilent 1100 series HPLC (Agilent Technologies Inc., Santa Clara, CA, USA). The temperature of the column was set at 35 °C, and the injection volume was 10 µL.

#### 4.2.4. Ultraviolet Spectra

A pumpkin polysacchrides solution (1mg/mL) was analyzed by ultraviolet spectroscopy on a UV-2401 ultraviolet spectrophotometer (Shimadzu Corporation, Kyoto, Japan) at wavelengths in the range of 200–800 nm.

#### 4.2.5. Fourier-Transform Infrared Spectra

Each sample was analyzed by Fourier-transform infrared spectroscopy on a VECTOR 22 Fourier transform infrared spectrometer (Thermo Fisher Scientific, Waltham, MA USA). Dried samples were ground with KBr powder before being pressed into pellets for analysis at wave numbers from 4000 to 400 cm^−1^.

### 4.3. Animals

Male Kunming mice, 12~14 weeks old, and weighing 13~15 g, were purchased from the Laboratory Animal Center of the Academy of Military Medical Science, Beijing, China (No. SCXK 2014-0013). The animals were maintained at a temperature of 22 ± 2 °C, a relative humidity of 55 ± 5%, and a 12 h light/dark cycle. The experimental protocols were approved by the local ethics committee for animal experiments. All procedures were performed in accordance with the Health Services Guide of Tianjin Agricultural University for the Care and Use of Laboratory Animals.

### 4.4. Experimental Design

After one week of accommodation, the mice were randomly divided into two groups based on body weight: a control (C) group (*n* = 10) and a model group (*n* = 40). The mice in the C group were fed with a normal standard diet, while those in the model group were fed with a high-fat diet (HFD) (78.8 g/100 g commercial standard pellet diet, 10 g/100 g lard, 10 g/100 g custard powder, 1 g/100 g cholesterol, and 0.2 g/100 g sodium cholate). After four weeks, the mice in the model group were injected intraperitoneally with 35 mg/kg STZ daily for 3 consecutive days. Blood samples were withdrawn from tail veins of mice that had been fasted overnight after induction for one week. The diabetic mice with blood glucose levels that were higher than 12 mM were randomly divided into four groups of 10 animals each, as follows: (a) the diabetic model (DM) group, (b) the pumpkin polysaccharides (PP) group, in which the mice were orally gavaged with pumpkin polysaccharides (400 mg/kg/day body weight), (c) the puerarin (P) group, in which mice were orally gavaged with puerarin (200 mg/kg/day body weight), and (d) the pumpkin polysaccharides and puerarin (PPP) group, in which mice were orally gavaged with pumpkin polysaccharides (200 mg/kg/day body weight) and puerarin (100 mg/kg/day body weight). All groups had free access to water.

### 4.5. Oral Glucose Tolerance Test

All mice were subjected to an OGTT after eight weeks of drug administration. The mice were fasted overnight and then glucose (2.5 g/kg body weight) was orally administered. The blood glucose levels were measured via blood obtained from tail veins of mice at 0, 30, 60, and 120 min after glucose administration. The results were expressed as the AUC over 120 min.

### 4.6. Evaluation of Blood Lipid Metabolism

At the end of the experimental period, the mice were anesthetized with pentobarbital sodium (70 mg/kg body weight) prior to sacrifice. Blood samples were collected from the retro-orbital vein and centrifuged at 2000 g for 20 min to gather the serum. The lipid levels, including FFAs, TC, TG, HDL, and LDL, in serum were analyzed. The blood insulin levels were determined by ELISA.

### 4.7. Oxidative Stress Assay

Liver tissues were homogenized in ice-cold phosphate-buffered saline and centrifuged at 1000 g for 10 min. The supernatant was collected for a biochemical analysis, including the ROS, MDA, and GSH levels and the SOD activity, to evaluate the antioxidant capacity. All assays were performed according to the manufacturers’ standards and protocols.

### 4.8. Western Blot Analysis

Cells were lysed in RIPA lysis buffer (50 mM Tris, 150 mM NaCl, 1 mL/100 mL NP-40, 0.5 g/100 mL sodium deoxycholate, pH 7.4) at 4 °C for 30 min. After removing the cellular debris by centrifugation at 10,000 *g* for 20 min, nuclear and cytoplasmic extracts from the supernatant were obtained by using a nuclear/cytoplasmic isolation kit. The proteins were subjected to SDS-PAGE and transferred onto a polyvinylidene fluoride membrane. The membrane was blocked with 5 g/100 mL skim milk for 2 h followed by incubation with the primary antibodies at 4 °C overnight. After washing with tris-buffered saline tween (TBST, 10 mM Tris-HCl, 150 mM Nacl, 0.5 mL/100 mL Tween 20, pH 7.4), the membranes were incubated with either HRP conjugated goat anti-mouse or goat anti-rabbit secondary antibody in TBST at 37 °C for 2 h. The bands were visualized using scanning gel electrophoresis and the enhanced chemilumescent detection system after washing the polyvinylidene fluoride membranes, and the density of the radiographic bands was analyzed.

### 4.9. Statistical Analyses

All values are expressed as means ± standard deviation of three replicates. The statistical analysis was carried out by using one-way ANOVA. The differences in the mean were calculated using Duncan’s multiple range test for means with a 95% confidence limit (*p* < 0.05).

## 5. Conclusions

The pumpkin polysaccharides and puerarin exhibited a potent glucose tolerance effect, and treatment with them could result in a decrease in blood glucose. In addition, they efficiently alleviated insulin resistance and exerted a cytoprotective action on T2DM mice. At the molecular level, they could upregulate the expression of the critical proteins in the Nrf2/HO-1 and PI3K/Akt signalling pathways.

## Figures and Tables

**Figure 1 molecules-24-00955-f001:**
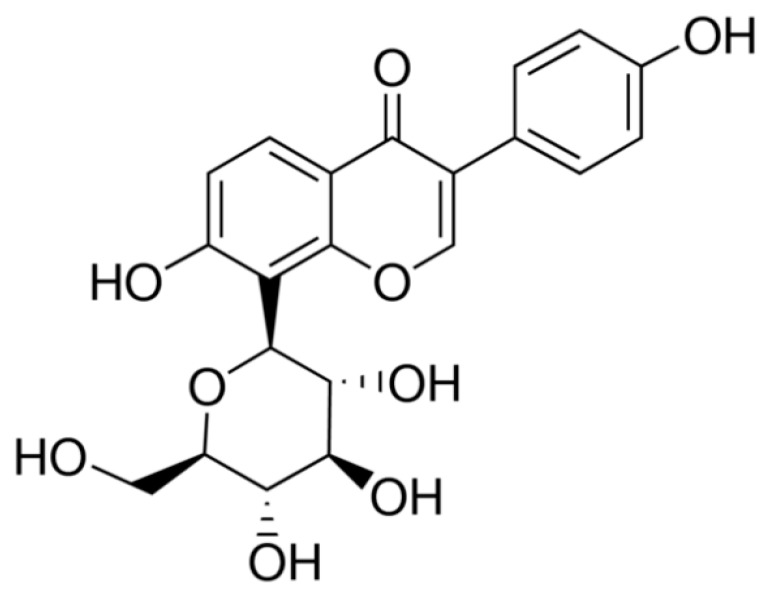
The chemical structure of puerarin.

**Figure 2 molecules-24-00955-f002:**
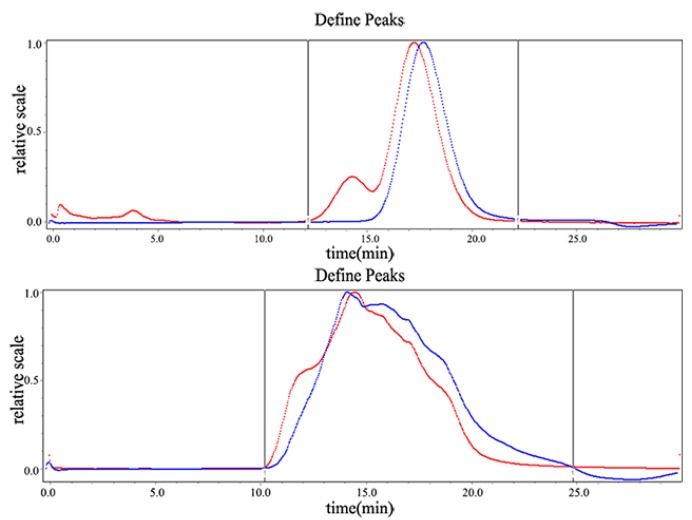
The molecular weight determination spectrum of pumpkin polysaccharides by HPGPC after the calibration of the dextran standard. The upper panel shows the dextran control, and the lower panel shows the pumpkin polysaccharides. The red lines represent the detection results of the 18-angle laser light scatterer, and the blue lines represent the detection results of the differential refractive index detector. The curve is the molecular weight distribution curve of the computer’s automatic statistics.

**Figure 3 molecules-24-00955-f003:**
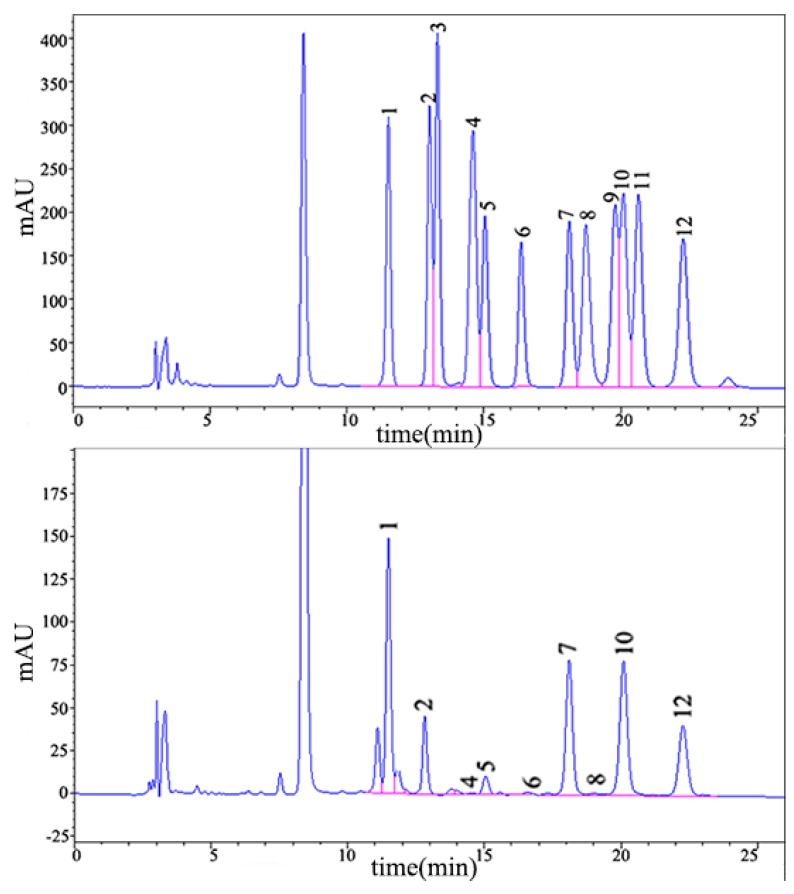
The monosaccharide composition analysis of the pumpkin polysaccharides by HPLC. The upper panel shows the monosaccharide control, and the lower panel shows the pumpkin polysaccharides. 1. mannose, 2. ribose, 3. rhamnose, 4. glucosamine, 5. glucuronic acid, 6. galacturonic acid, 7. glucose, 8. galactosamine, 9. galactose, 10. xylose, 11. arabia sugar, and 12. fucose.

**Figure 4 molecules-24-00955-f004:**
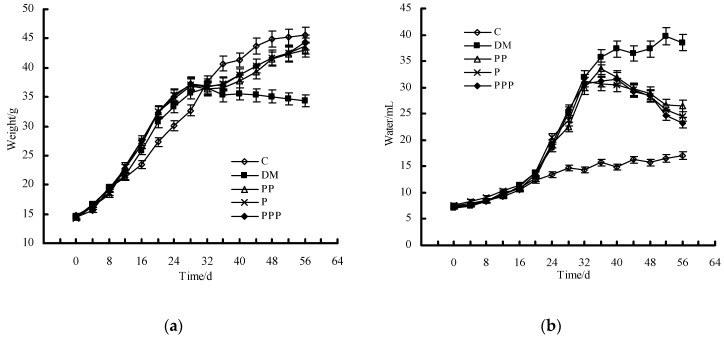
The effect of pumpkin polysaccharides and puerarin on the body weight and water intake of mice (*n* = 10). (**a**) The effect of pumpkin polysaccharides and puerarin on the body weight. (**b**) The effect of pumpkin polysaccharides and puerarin on the water intake. C, control; DM, diabetic model; PP, pumpkin polysaccharides; P, puerarin; PPP, pumpkin polysaccharides and puerarin.

**Figure 5 molecules-24-00955-f005:**
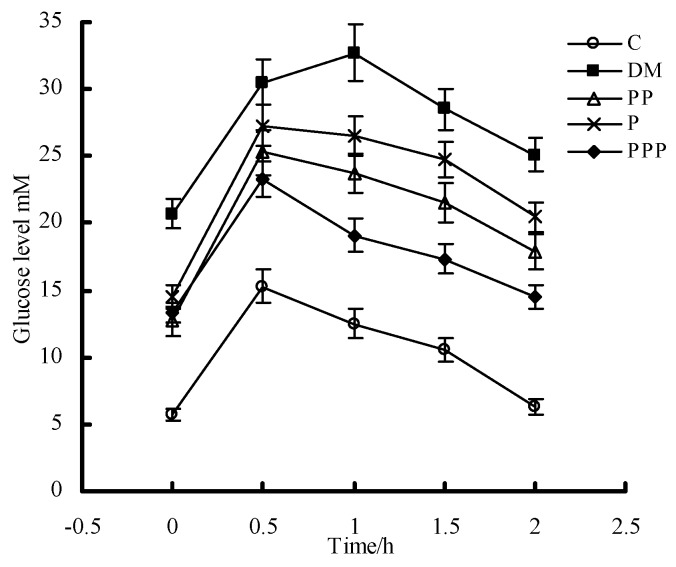
The oral glucose tolerance test of pumpkin polysaccharides and puerarin (*n* = 10). C, control; DM, diabetic model; PP, pumpkin polysaccharides; P, puerarin; PPP, pumpkin polysaccharides and puerarin.

**Figure 6 molecules-24-00955-f006:**
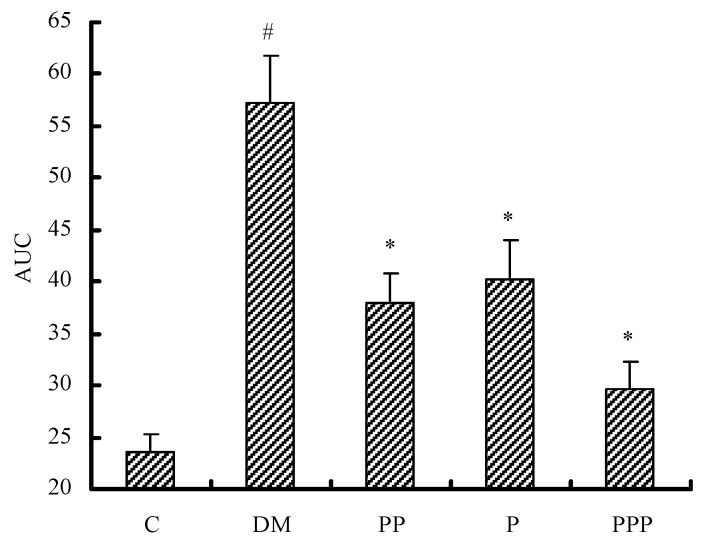
The effect of pumpkin polysaccharides and puerarin on the AUC of glucose tolerance (*n* = 10). C, control; DM, diabetic model; PP, pumpkin polysaccharides; P, puerarin; PPP, pumpkin polysaccharides and puerarin. ^#^
*p* < 0.05 compared with the C group; * *p* < 0.05 compared with the DM group.

**Figure 7 molecules-24-00955-f007:**
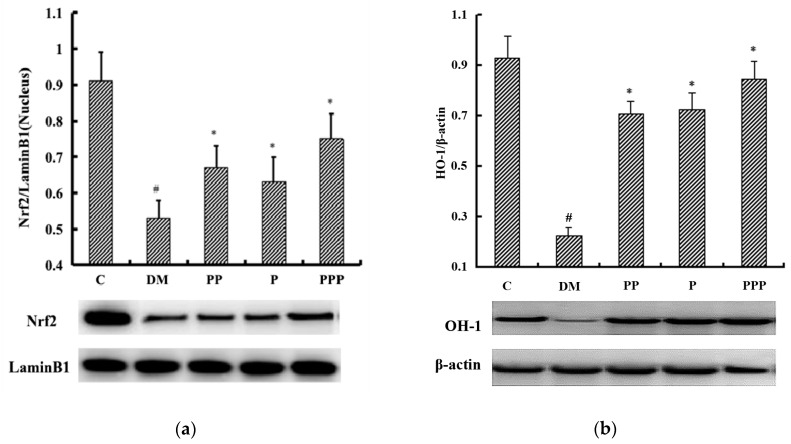
A Western blot analysis of the proteins that are associated with the Nrf2/HO-1 pathway (*n* = 10). (**a**) A relative density analysis of Nrf2 in the nucleus. (**b**) A relative density analysis of HO-1. The relative density of the Nrf2 and HO-1 protein bands is expressed as a ratio (Nrf2/Lamin B1, HO-1/β-actin). ^#^
*p* < 0.05 compared with the C group; * *p* < 0.05 compared with the DM group.

**Figure 8 molecules-24-00955-f008:**
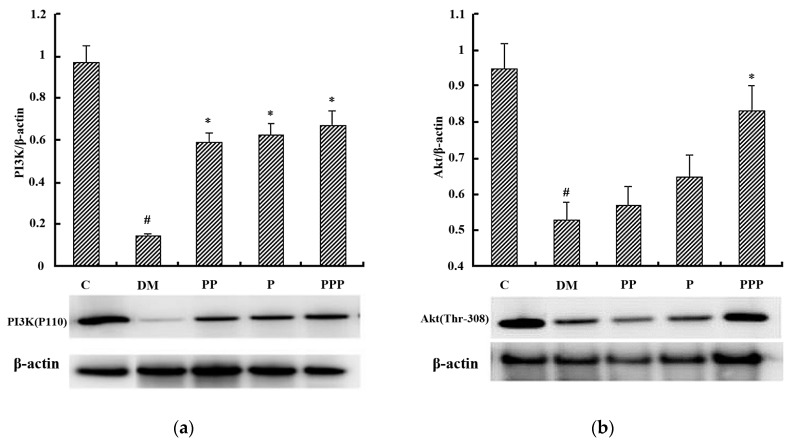
A Western blot analysis of the proteins that are associated with the PI3K/Akt pathway (*n* = 10). (**a**) A relative density analysis of PI3K. (**b**) A relative density analysis of Akt. The relative density of PI3K and Akt is expressed as a ratio (PI3K/β-actin, Akt/β-actin). ^#^
*p* < 0.05 compared with the C group; * *p* < 0.05 compared with the DM group.

**Table 1 molecules-24-00955-t001:** The effect on the serum glucose and insulin level.

Group	Fasting Blood Glucose (mM)	Insulin (mU/L)	IR
C	3.03 ± 0.27	6.78 ± 0.59	1.11 ± 0.09
DM	17.85 ± 1.53 ^#^	13.05 ± 0.95 ^#^	10.77 ± 1.05 ^#^
PP	9.87 ± 1.07 *	8.37 ± 0.75 *	3.75 ± 0.35 *
P	11.23 ± 1.28 *	8.03 ± 0.66 *	4.05 ± 0.47 *
PPP	6.35 ± 0.53 *	7.05± 0.57 *	2.53 ± 0.23 *

C, control; DM, diabetic model; PP, pumpkin polysaccharides; P, puerarin; PPP, pumpkin polysaccharides and puerarin; IR, insulin resistance. *n* = 10; ^#^
*p* < 0.05 compared with the C group; * *p* < 0.05 compared with the DM group.

**Table 2 molecules-24-00955-t002:** The effect on serum lipid levels.

Group	TG (mM)	TC (mM)	HDL (mM)	LDL (mM)	FFAs (µM)
C	0.87 ± 0.09	2.85 ± 0.23	1.75 ± 0.13	0.57 ± 0.05	0.37 ± 0.03
DM	2.75 ± 0.23 ^#^	5.28 ± 0.49 ^#^	0.83 ± 0.07 ^#^	2.79 ± 0.23 ^#^	0.95 ± 0.09 ^#^
PP	1.73 ± 0.15 *	4.02 ± 0.37 *	1.33 ± 0.11 *	1.73 ± 0.15 *	0.53 ± 0.05 *
P	1.57 ± 0.13 *	3.79 ± 0.35 *	1.41 ± 0.13 *	1.57 ± 0.13 *	0.47 ± 0.05 *
PPP	1.18 ± 0.11 *	3.18 ± 0.31 *	1.68 ± 0.13 *	1.22 ± 0.11 *	0.39 ± 0.03 *

C, control; DM, diabetic model; PP, pumpkin polysaccharides; P, puerarin; PPP, pumpkin polysaccharides and puerarin. *n* = 10; ^#^
*p* < 0.05 compared with the C group; * *p* < 0.05 compared with the DM group.

**Table 3 molecules-24-00955-t003:** The effect on oxidative stress.

Group	SOD (U/mg)	GSH (U/mg prot)	MDA (nmol/L)	ROS (IU/L)
C	217.95 ± 19.09	27.57 ± 3.09	6.07 ± 0.47	509.3 ± 21.93
DM	133.59 ± 10.65 ^#^	15.77 ± 1.31 ^#^	11.59 ± 0.95 ^#^	597.1 ± 25.01 ^#^
PP	175.47 ± 11.05 *	21.85 ± 1.75 *	7.31 ± 0.66 *	537.5 ± 20.87 *
P	191.33 ± 13.03 *	21.37 ± 1.91 *	7.07 ± 0.59 *	531.5 ± 23.07 *
PPP	209.11 ± 17.85 *	25.07 ± 2.53 *	6.28 ± 0.53 *	517.3 ± 21.55 *

C, control; DM, diabetic model; PP, pumpkin polysaccharides; P, puerarin; PPP, pumpkin polysaccharides and puerarin. *n* = 10; ^#^
*p* < 0.05 compared with the C group; * *p* < 0.05 compared with the DM group.

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
