# Peer review of "Synergistic Hypoglycemic Effects of Pumpkin Polysaccharides and Puerarin on Type II Diabetes Mellitus Mice"

_molecules, 2019, doi:10.3390/molecules24050955_

Round 1

Reviewer 1 Report

This is an interesting work dealing with the characterization of pumkin polysaccharides and their antidiabetic effec in an animal experimental model. 

Generally, the study is well assessed and quite comprehensive as it enters also in the mechanistic approaches, trying to better explain the mechanisms behind the pharmacological action of the polysaccharides. 

Besides this paper is further considered, I would have some comments: 

language style and overall style should be slightly improved as well as reference style; 

does this study has an Ethical commission approval? This has to be mentioned in the manuscipt clearly ... 

Author Response

Point 1: language style and overall style should be slightly improved as well as reference style 

Response 1: As MDPI provides an English editing service checking grammar, spelling, punctuation and some improvement of style where necessary, according to the reviewer and the academic editors comments, we have our manuscript checked by using the professional English editing service.

Point 2: does this study has an Ethical commission approval? This has to be mentioned in the manuscipt clearly ... 

Response 2: Our manuscript contains information related to animal use, and our study need ethic approval. According to the editor’s comment, we provided scanned copy of ethic approval files when submitting the manuscript. 

Reviewer 2 Report

The manuscript be Xue Chen, Lei Qian, Bujiang Wang, Zhijun Zhang, Han Liu, Yeni Zhang and Jinfu Liu titled "Synergistic Hypoglycemic Effects of Pumpkin Polysaccharides and Puerarin in Type II Diabetes Mellitus Mice" presents results on mice of extracts of pumpkin and puerarin on glucose tolerance, water consumption, body weight, and some biological pathways on a mouse model of diabetes. The following issues arise:

1) There are a lot of published work on puerarin and on pumpkin extracts, as related to research in the field of diabetes. As such, there is not a high level of novelty of the present work.

2) The purported synergistic effect of puerarin and pumpkin extract is small, almost indistinguishable in body weight and water consumption.

3) The figures can be improved since the legends and axis and labels are not clearly legible.

Author Response

Point 1: There are a lot of published work on puerarin and on pumpkin extracts, as related to research in the field of diabetes. As such, there is not a high level of novelty of the present work.

Response 1: Many researchers have devoted themselves to the study of polysaccharides; so far, only acidic polysaccharides have been found in pumpkin (Zhu et al, 2009; Zhao et al, 2017). Pumpkin acidic polysaccharides are mainly composed of rhamnose, galacturonic acid, galactose and arabinose, with a molecular ratio of 7.4: 25: 28: 2.6. Based on the identified characteristics, the pumpkin polysaccharides that have been separated in our lab are novel.

Zhu, F.; Fan, W.; Zhu, Y.; Li X. Properties and analysis of pumpkin polysaccharide by UV and infrared spectrometries. Chinese Journal of Spectroscopy Laboratory 2009, 3, 609-612. 

Zhao, J. Structure elucidation of pumpkin acidic polysaccharides and the interaction with functional proteins. Doctoral Dissertation of China Agricultural University 2017.

Although the anti-diabetic effects of puerarin and pumpkin polysaccharides in rodent diabetes models have already been described in multiple studies, their synergistic hypoglycemic effect has not been reported. In the present study, we verified the synergistic hypoglycemic effect of pumpkin polysaccharide and puerarin coadministration on T2DM mice, and preliminary investigated the potential mechanism.

Point 2: The purported synergistic effect of puerarin and pumpkin extract is small, almost indistinguishable in body weight and water consumption.

Response 2: Actually the synergistic effect of puerarin and pumpkin extract is small in body weight and water consumption. At the same time in some aspects there are no significant difference between the single groups and the combined group, especially the serum insulin level and ROS level.

However, in most aspects the combined administration of pumpkin polysaccharides and puerarin could synergistically ameliorate T2DM, such as lowering the glucose level and the insulin resistance value, reducing the blood glucose response, decreasing the serum glucose level, reducing serum TG and LDL levels and the content of free fatty acids, improving serum HDL level and increasing GSH level. On all aspects mentioned above the synergetic effect is better, and there are significant difference between the single groups and the combined group. Therefore in general the synergetic effect of puerarin and pumpkin polysaccharides is obvious.

Point 3: The figures can be improved since the legends and axis and labels are not clearly legible.

Response 3: According to the reviewer’s comment, to make the legends and axis and labels clearly legible, we provided the figures again. 

Reviewer 3 Report

COMMENTS:

The manuscript entitled “Synergistic Hypoglycemic Effects of Pumpkin Polysaccharides and Puerarin in Type II Diabetes Mellitus Mice” examined that the effects of pumpkin polysaccharides and puerarin against impairing diabetes of mice that were treated with both high fat diet and low dose streptozotocin injection. The pumpkin polysaccharides were purified and the structure was estimated. Administrations of the purified pumpkin polysaccharides and puerarin were effective to inhibit worsening several biomarkers in the mice. Furthermore, the phosphorylation of the proteins related to insulin signal pathways were altered by administrations of pumpkin polysaccharides and puerarin.

The study provides important information to use the pumpkin polysaccharides and puerarin for treatment of diabetes.

Comment:

1). What are criteria to determine “synergistic hypoglycemic effects”? It is possible that the effects are additive, not synergistic. FIC (fractional inhibitory concentration) index is used to determine synergistic or not in the field of antibiotics.

2). Streptozotocin induces onset of Type I diabetes by destruction of insulin producing cells. In this study, low dose (35 mg/kg) of streptozotocin was administered to mice for onset of diabetes. The authors described that the diabetic mice used in this study closely mimic the metabolic characteristics of human T2DM in discussion section (page 10, line 227-229). How did the similarity of the characteristics between the mice used in this study and human T2DM determine? Is it correct to describe “Type II diabetes mellitus mice”?

3). How did the authors obtain the pueranin? What is purity of the pueranin used in this study?

4). I cannot understand the meanings of the upper panel and lower panel in Figure 2. In Figure 3, upper panel probably shows monosaccharide control and lower panel shows sample of pumpkin polysaccharides. Further explanations are needed to understand what are those. The vertical axis in Figure 3, 4, and 5 should be added. The word size in Figure 2, 3, 4, and 5 should be bigger, because it is difficult to see.

5). References should be cited end of the sentences (page 8, line 188-189, page 8, line 189-191, and page 9, line 210-211).

Author Response

Point 1: What are criteria to determine “synergistic hypoglycemic effects”? It is possible that the effects are additive, not synergistic. FIC (fractional inhibitory concentration) index is used to determine synergistic or not in the field of antibiotics.

Response 1: The Chou-Talalay method for drug combination is based on the median-effect equation, derived from the mass-action law principle, which is the unified theory that provides the common link between single entities, and first order and higher order dynamics. The resulting combination index (CI) theorem of Chou-Talalay offers quantitative definition for additive effect (CI=1), synergism (CI<1), and antagonism (CI>1) in drug combinations.

To determine whether the hypoglycemic effects of pumpkin polysaccharides and puerarin is synergistic or additive, we calculated the combined index (CI) using the medium-effect principle of Chou and Talady. CI=D1/Dx1+D2/Dx2 (D1 and D2 are the action concentration of drug 1 and drug 2 when the combination inhibition rate is 50%, Dx1 and Dx2 are the action concentration of drug 1 and drug 2 when the separate inhibition rate is 50%). If CI=1, the combined effect has a superposition effect; CI<1, the combined effect has synergistic effect; CI>1, the combined effect has antagonistic effect.

Chou, T.C. Theoretical basis, experimental design, and computerized simulation of synergism and antagonism in drug combination studies. Pharmacol. Rev. 2006, 58, 621-681.

Chou, T.C. Drug combination studies and their synergy quantification using the Chou -Talalay method. Cancer Res. 2010, 70, 440-446.

Point 2: Streptozotocin induces onset of Type I diabetes by destruction of insulin producing cells. In this study, low dose (35 mg/kg) of streptozotocin was administered to mice for onset of diabetes. The authors described that the diabetic mice used in this study closely mimic the metabolic characteristics of human T2DM in discussion section (page 10, line 227-229). How did the similarity of the characteristics between the mice used in this study and human T2DM determine? Is it correct to describe “Type II diabetes mellitus mice”?

Response 2: Animal models mimicking human T2DM are pivotal for the biological screening of anti-diabetic drugs, and several approaches have been used to develop such models. A good way of initiating the insulin resistance associated with T2DM in humans is the administration of HFD to animals, thus inducing obesity, which acts as a known risk factor for T2DM. Several animal models have been employed in the diabetes research out of which the HFD regimen was utilized to induce T2DM (Kahn et al, Nature 2006; Nath et al, J. Pharmacol. Tox. Met. 2017; Chandrasekaran et al, Biomed. Pharmacother. 2018; Nasimeh et al, Adv. Biomed. Res. 2018). In this study, the animal model for T2DM was induced by feeding the animals with HFD following low dose STZ. However, the sentence "which would closely mimic the metabolic characteristics of human T2DM" needs to be rephrased. According to the reviewer’s comment, to reduce ambiguity, the sentence was deleted.

Kahn, S. E.Hull, R. L.Utzschneider, K. M. Mechanisms linking obesity to insulin resistance and type 2 diabetes. Nature 2006, 444, 840–846. 

Nath, S.; Ghosh, S. K.; Choudhury, Y. A murine model of type 2 diabetes mellitus developed using a combination of high fat diet and multiple low doses of streptozotocin treatment mimics the metabolic characteristics of type 2 diabetes mellitus in humans. J. Pharmacol. Tox. Met. 2017, 84, 20-30.

Chandrasekaran, S.; Ramajayam, N.; Pachaiappan, P. Ameliorating effect of berbamine on hepatic key enzymes of carbohydrate metabolism in high-fat diet and streptozotocin induced type 2 diabetic rats. Biomed. Pharmacother. 2018, 103, 539-545.

Nasimeh, V.; Farzaneh, R.; Ahmad, R. S.; Sharifeh, K.; Gholamreza, D.; Gilda, E.; Sedigheh, M.; Rasoul, S. Novel high-fat diet formulation and streptozotocin treatment for induction of prediabetes and type 2 diabetes in rats. Adv. Biomed. Res. 2018, 7, 107.

Point 3: How did the authors obtain the pueranin? What is purity of the pueranin used in this study?

Response 3: Puerarin was purchased from Sigma Chemical Co. (St. Louis, MO, USA) and the purity of the pueranin used in this study is over 98%. 

Point 4: I cannot understand the meanings of the upper panel and lower panel in Figure 2. In Figure 3, upper panel probably shows monosaccharide control and lower panel shows sample of pumpkin polysaccharides. Further explanations are needed to understand what are those. The vertical axis in Figure 3, 4, and 5 should be added. The word size in Figure 2, 3, 4, and 5 should be bigger, because it is difficult to see.

Response 4: In Figure 2, the upper panel shows the dextran control, and the lower panel shows the pumpkin polysaccharides. Molecular weight was evaluated by HPGPC. The red lines represent the detection results of the 18-angle laser light scatter, and the blue lines represent the detection results of the differential refractive index detector. The curve is the molecular weight distribution curve of the computer’s automatic statistics.

In Figure 3, the upper panel shows the monosaccharide control, and the lower panel shows the pumpkin polysaccharides. The vertical axis in Figure 3 is mAU.

According to the academic editor’s comment, we removed Figure 4 and 5. According to the reviewer’s comment, we adjusted the word size in Figure 2 and 3.

Point 5: References should be cited end of the sentences (page 8, line 188-189, page 8, line 189-191, and page 9, line 210-211).

Response 5: According to the reviewer’s comment, the references were provided. 

Round 2

Reviewer 1 Report

No further comments. Only language style and overall style should be improved during proof if accepted. 

Author Response

Point 1: No further comments. Only language style and overall style should be improved during proof if accepted.

Response 1: Thanks to the reviewer’s comments. The language style and overall style will be improved. 

Reviewer 2 Report

I still believe the points raised in previous rounds of review have not been overcome. The preset manuscript does not present strong synergy or sufficient novelty for publication.

Author Response

Point 1: I still believe the points raised in previous rounds of review have not been overcome. The preset manuscript does not present strong synergy or sufficient novelty for publication.

Response 1: The anti-diabetic effects of pumpkin polysaccharides in rodent diabetes models have been described in some pieces of literature. 

Xiong et al proved that polysaccharide from pumpkins had remarkably better effect of reducing glycemia on alloxan induced diabetic rats (Xiong et al, Chinese Journal of Modern Applied Pharmacy, 2001).

Zhang et al found that different kinds of pumpkin polysaccharides have different hypoglycaemic effects and this was exemplified using alloxan-induced diabetic rats (Zhang et al, Food and Fermentation Industries, 2002).

To investigate hypoglycemic substances from pumpkin, protein-bound polysaccharide (PBPP) was isolated from water soluble substances of the fruits (Li et al, Plant Food Hum Nutr, 2005). PBPP can obviously increase the levels of serum insulin, reduce the blood glucose levels and improve tolerance of glucose in alloxan diabetic rats. 

The polysaccharides from pumpkin fruit (PP) protected islets cells from STZ injury in vitro via increasing the levels of SOD and MDA and reducing the production of NO (Zhu et al, Chin J Nat Medicines, 2015). The molecular weight of the polysaccharides from pumpkin fruit was about 23 kDa and PP was composed of Arabinose, Mannose, Glucose, and Galactose with a molar ratio of 1:7.79:70.32:7.05.

Other researchers have devoted themselves to the study of pumpkin polysaccharides; so far, only acidic polysaccharides have been found (Zhu et al, Chinese Journal of Spectroscopy Laboratory, 2009; Zhao, Doctoral Dissertation of China Agricultural Universityl, 2017). Pumpkin acidic polysaccharides are mainly composed of rhamnose, galacturonic acid, galactose and arabinose, with a molecular ratio of 7.4: 25: 28: 2.6.

Polysaccharides are the bioactive materials of pumpkin. More detailed work should be undertaken in order to find some clues on the relationship between the differences in compositions and hypoglycaemic activity of polysaccharides. 

In this manuscript the pumpkin polysacchrides are mainly composed of mannose, ribose, glucose, xylose and fucose, with a molecular weight of 749.3 kDa. Based on the identified characteristics, the pumpkin polysaccharides are novel.  

There are some published work on puerarin, as related to research in the field of diabetes. The antihyperglycemic action of puerarin was investigated in STZ-diabetic rats (Hsu et al, J Nat Prod, 2003). Puerarin significantly attenuated the increase of plasma glucose. In the isolated soleus muscle of STZ-diabetic rats, puerarin enhanced the uptake of radioactive glucose. Moreover, the mRNA and protein levels of GLUT4 in soleus muscle were increased.

Li et al found that puerarin protects pancreatic b-cell function and survival via direct effects on b-cells, and its protection of b-cell survival is mediated by the PI3K/Akt pathway (Li et al, J Mol Endocrinol, 2014).

Puerarin improved glucose homeostasis in obese diabetic mice and identified a novel role of puerarin in protecting β-cell survival by mechanisms involving activation of GLP-1R signaling and downstream targets (Yang et al, Mol Endocrinol, 2016).

Numerous studies have shown that active ingredients, including flavonoids, polyphenols, alkaloids, and polysaccharides, such as pumpkin, bean, oat, and balsam pear polysaccharides, exert beneficial effects on DM, and the combination of two or more kinds of ingredients can usually achieve better results through different targets.

Although the anti-diabetic effects of puerarin and pumpkin polysaccharides in rodent diabetes models have already been described in some studies, their combined hypoglycemic effect has not been reported. To elucidate the mechanism that underlies the compatibility between pumpkin polysaccharides and puerarin, in the present study we verified the synergistic hypoglycemic effect of pumpkin polysaccharide and puerarin coadministration on T2DM mice, and preliminary investigated the potential mechanism. 

In this manuscript both the single groups and the combined group could ameliorate T2DM, and the combined hypoglycemic effects are better. Actually the synergistic effect of puerarin and pumpkin polysaccharides is small in body weight and water consumption, while in some aspects there are no significant difference between the single groups and the combined group, especially the ROS level (p>0.05).

However, in some aspects the combined administration of pumpkin polysaccharides and puerarin could significantly ameliorate T2DM, such as lowering the glucose level, decreasing the fasting blood glucose level and the insulin resistance value, reducing serum lipid levels and the content of free fatty acids (p<0.01) (Table 1). In other aspects there are also difference between the single groups and the combined group, including lowering the insulin level, reducing the content of MDA, increasing the SOD and GSH levels (p<0.05) (Table 1). Therefore in general the combined effect of puerarin and pumpkin polysaccharides is obvious.

Table 1

PP

P

PPP

Descriptions

Glucose level

Initial content (mM)

12.7 ± 1.1

14.5 ± 0.9

13.3 ± 0.7

Final content (mM)

17.9 ± 1.3

20.5 ± 1.1

14.5 ± 0.9

Increment (%)

40.95 **

41.38 **

9.03

strong synergism

Glucose tolerance

AUC

37.9 ± 2.9 **

40.3 ± 3.7 **

29.7 ± 2.5

moderate synergism

Serum glucose

Insulin level

Fasting blood glucose (mM)

9.87 ± 1.07 **

11.23 ± 1.28 **

6.35 ± 0.53

common synergism

Insulin (mU/L)

8.37 ± 0.75 **

8.03 ± 0.66 *

7.05± 0.57

slight synergism

Insulin resistance

3.75 ± 0.35 **

4.05 ± 0.47 **

2.53 ± 0.23

common synergism

Serum lipid levels

TG (mM)

1.73 ± 0.15 **

1.57 ± 0.13 **

1.18 ± 0.11

common synergism

TC (mM)

4.02 ± 0.37 **

3.79 ± 0.35 **

3.18 ± 0.31

moderate synergism

HDL (mM)

1.33 ± 0.11 **

1.41 ± 0.13 **

1.68 ± 0.13

moderate synergism

LDL (mM)

1.73 ± 0.15 **

1.57 ± 0.13 **

1.22 ± 0.11

moderate synergism

FFAs (µM)

0.53 ± 0.05 **

0.47 ± 0.05 **

0.39 ± 0.03

moderate synergism

Oxidative stress

SOD (U/mg)

175.47 ± 11.05 **

191.33 ± 13.03 *

209.11 ± 17.85

slight synergism

GSH (U/mg prot)

21.85 ± 1.75 *

21.37 ± 1.91 *

25.07 ± 2.53

moderate synergism

MDA (nmol/L)

7.31 ± 0.66 **

7.07 ± 0.59 *

6.28 ± 0.53

slight synergism

PP, pumpkin polysaccharides; P, puerarin; PPP, pumpkin polysaccharides and puerarin; *p <0.05 and **p <0.01 compared with the PPP group.

Descriptions of the CI values was graded as very strong synergism, strong synergism, common synergism, moderate synergism, slight synergism, etc. For example, the combination of pumpkin polysaccharides at 200 mg/kg and puerarin at 100 mg/kg caused 9.03% increment (from 13.3±0.7 mM to 14.5±0.9 mM) of the glucose level, which is lower than the increment caused by pumpkin polysaccharides (40.95% increment, from 12.7±1.1 mM to 17.9±1.3 mM) at 400 mg/kg or puerarin (41.38% increment, from 14.5±0.9 mM to 20.5±1.1 mM) at 200 mg/kg (twice the concentration of each drug). The CI value suggests a strong synergistic interaction between pumpkin polysaccharides and puerarin. On the other aspects mentioned above, the CI values for the combinations of pumpkin polysaccharides with puerarin suggest common synergism, moderate synergism, slight synergism, respectively (Table 1).

As the pumpkin polysacchrides are novel and the combined hypoglycemic effect with puerarin has not been reported, the preset manuscript presents sufficient novelty. Although the pumpkin polysaccharides and puerarin combination does not present strong synergy in all aspects, actually they has a synergetic hypoglycemic effect.

Reviewer 3 Report

The authors partially responded to my comments. 

I have understood that the authors used a method to determine synergistic or additive effect of a combination of chemicals by combination index (CI). However, the revised manuscript was not explained about CI of pumpkin polysaccharides and puerarin combination. What is the result of the CI? The CI is less than 1? The result is important because if the CI is not less than 1, the title of this manuscript should be changed.

Author Response

Point 1: The authors partially responded to my comments. I have understood that the authors used a method to determine synergistic or additive effect of a combination of chemicals by combination index (CI). However, the revised manuscript was not explained about CI of pumpkin polysaccharides and puerarin combination. What is the result of the CI? The CI is less than 1? The result is important because if the CI is not less than 1, the title of this manuscript should be changed.

Response 1: Insulin resistance is the major factor leading to T2DM (Minshall et al, Clin. Ther., 2005; Jia, New England Journal of Medicine, 2010). IR HepG2 cell model could accurately mimic the physiological state of T2DM patients, and it has become an ideal model for screening functional components and studying the mechanism to ameliorate T2DM in vitro (KUMAR, Philadelphia, 2005; Gupta et al, J Cell Biochem, 2007).

In our previous study, IR HepG2 cell model was developed after insulin incubation. Treatment with the combination of pumpkin polysaccharides and puerarin significantly increased consumption of extracellular glucoses compared with individual drugs.

As the degree of synergism in combination treatments can vary significantly with the drug ratio, a 7×7 Latin Square design was constructed with seven different constant molar ratios (from 8:1 to 1:8) between the two drugs. Analysis of dose–effect relationships was performed according to the median-effect method of Chou and Talalay and CI values were calculated. In the puerarin to pumpkin polysaccharides ratios of 1:2, there was maximum synergy in vitro. The combinations showed a strong synergism at low concentrations gradually changing to common synergism at high concentrations.

According to the results in vitro (not published), in this manuscript the mice in the puerarin and pumpkin polysaccharides group were gavaged with puerarin (200 mg/kg/day body weight) and pumpkin polysaccharides (400 mg/kg/day body weight), respectively. The mice were gavaged with puerarin (100 mg/kg/day body weight) and pumpkin polysaccharides (200 mg/kg/day body weight) in the combined group (half the concentration of each drug).

Both pumpkin polysaccharides and puerarin could ameliorate T2DM, and the pumpkin polysaccharides and puerarin combination has a synergetic hypoglycemic effect.

In some aspects the combined administration significantly ameliorate T2DM, such as lowering the glucose level, decreasing the fasting blood glucose level and the insulin resistance value, reducing serum lipid levels and the content of free fatty acids (p<0.01).

For example, the combination of pumpkin polysaccharides at 200 mg/kg and puerarin at 100 mg/kg caused 9.03% increment (from 13.3±0.7 mM to 14.5±0.9 mM) of the glucose level, which is lower than the increment caused by pumpkin polysaccharides (40.95% increment, from 12.7±1.1 mM to 17.9±1.3 mM) at 400 mg/kg or puerarin (41.38% increment, from 14.5±0.9 mM to 20.5±1.1 mM) at 200 mg/kg (twice the concentration of each drug). The CI value is less than 1.0, suggesting a synergistic interaction between pumpkin polysaccharides and puerarin.

In other aspects there are also difference between the single groups and the combined group, including lowering the insulin level, reducing the content of MDA, increasing the SOD and GSH levels (p<0.05).

On all aspects mentioned above the synergetic effect is better, and the CI values for the combinations of pumpkin polysaccharides with puerarin are less than 1.0, which suggests synergism.